# Using FIBexDB for In-Depth Analysis of Flax Lectin Gene Expression in Response to *Fusarium oxysporum* Infection

**DOI:** 10.3390/plants11020163

**Published:** 2022-01-07

**Authors:** Natalia Petrova, Natalia Mokshina

**Affiliations:** Kazan Institute of Biochemistry and Biophysics, FRC Kazan Scientific Center of RAS, Lobachevsky Str., 2/31, 420111 Kazan, Russia; npetrova@inbox.ru

**Keywords:** flax (*Linum usitatissimum*), *Fusarium oxysporum*, FIBexDB, plant lectins, gene expression, coexpression network

## Abstract

Plant proteins with lectin domains play an essential role in plant immunity modulation, but among a plurality of lectins recruited by plants, only a few members have been functionally characterized. For the analysis of flax lectin gene expression, we used FIBexDB, which includes an efficient algorithm for flax gene expression analysis combining gene clustering and coexpression network analysis. We analyzed the lectin gene expression in various flax tissues, including root tips infected with *Fusarium oxysporum*. Two pools of lectin genes were revealed: downregulated and upregulated during the infection. Lectins with suppressed gene expression are associated with protein biosynthesis (Calreticulin family), cell wall biosynthesis (galactose-binding lectin family) and cytoskeleton functioning (Malectin family). Among the upregulated lectin genes were those encoding lectins from the Hevein, Nictaba, and GNA families. The main participants from each group are discussed. A list of lectin genes, the expression of which can determine the resistance of flax, is proposed, for example, the genes encoding amaranthins. We demonstrate that FIBexDB is an efficient tool both for the visualization of data, and for searching for the general patterns of lectin genes that may play an essential role in normal plant development and defense.

## 1. Introduction

It is generally accepted that lectins are proteins capable of recognizing and binding specifically and reversibly to carbohydrate structures without changing the carbohydrate moiety. Initially, lectins were discovered as highly toxic molecules, and their protective function in plant life was considered to be directly entomotoxic [1]. Following the discovery of non-toxic lectins, a better understanding of the role of proteins with lectin domains in plant defense and symbiotic interactions has emerged; such participation occurs via signaling pathways which are associated with microbe- or pathogen-associated molecular patterns (PAMP and MAMP) [2,3]. At the same time, involvement of plant lectins in cell organization, embryo morphogenesis, phagocytosis, growth mechanisms of cells, induced mitosis, pollinic recognition, and active engagement in the transport of carbohydrates and their establishment in plant tissues has been demonstrated [4].

Based on different criteria, such as subcellular localization, molecular structure, sequence, or abundance, lectins can be classified in different ways [5]. Plant proteins with lectin domains are divided into 12–18 families in accordance with their conserved carbohydrate-binding sites, the sequence, and the three-dimensional structure of the lectin motif [5,6,7]. The majority of the families are named after their most studied representative within the group, for example, GNA (*Galanthus nivalis* agglutinin), Legume (first described for *Leguminosae*), Malectin and Malectin-like (maltose and related oligosaccharide binding lectin), LysM (Lysin motif), Nictaba (*Nicotiana tabacum* agglutinin), Calreticulin (calcium-binding protein present in the endoplasmic reticulum), etc. [6]. Analysis of fully sequenced plant genomes revealed a huge variety of plant lectins, both quantitative and qualitative. A huge proportion of the lectin genes found in the genome turned out to encode “inducible” proteins, the synthesis of which is switched on under certain influences. These studies have greatly expanded the functions of lectins in addition to the historically previously described direct insecticidal role, which is related to their interaction and binding to a variety of glycosylated targets in the insect gastrointestinal tract. An important feature of plant lectins is the presence of repeated combinations of certain protein domains, such as the GNA, Legume, LysM, Malectin, and Malectin-like domains, along with a protein kinase domain. Such chimeric proteins form an extensive group of lectin receptor-like kinases (LecRLK), which are part of the plant immune system [8]. LecRLKs in the formation of responses to pathogens and symbionts act as the initial stage in the signaling cascade, transmitting the signal about elicitors along the MAPK defense signaling cascade to cellular targets, for example, transcription factors that regulate the expression of protective genes [9].

Recently, the revision of flax (*Linum usitatissimum* L.) proteins with lectin domains has been carried out [6]; 406 genes encoding lectin domain-containing proteins in the flax genome were identified. It was reported that about 75% of all these proteins were expressed in different tissues of the flax stem during normal plant development. Different sets of upregulated genes were found in tissues enriched with primary, secondary, and tertiary cell walls [6]. In the current work, we consider flax lectins via the prism of their “classic” function (as defense proteins) and analyze the behavior of flax lectins in root tips during fungal infection. The transcriptomic data for different flax cultivars’ root tips, which reflected the early response of flax plants to *Fusarium oxysporum* action [10], was integrated into the FIBexDB [11,12], and we were able to assess the response of stage-specific genes encoding proteins with lectin domains in the case of Fusarium wilt of flax plants and compare them to samples attributed to stem tissues of plants grown under normal conditions.

Fusarium wilt is a nasty and common disease affecting economically important crops, including flax. Flax response to *Fusarium oxysporum* has been described previously by other researchers from the transcriptomic point of view [10,13,14], but no one had studied the expression of flax genes encoding lectin-domain containing proteins and no one had considered coexpression networks for lectin genes using the flax transcriptome database, while proteins with lectin domains could play an essential role in plant immunity modulation and determine the resistance of flax plants. Establishing the function of inducible lectins and their roles in the signaling pathways in which they engage is not only fundamental research, but it can also have a significant impact on agricultural applications. The inclusion of lectins will surely aid the development of more stress-resistant crops using traditional breeding methods or recombinant technology. In the present paper, we propose an effective algorithm for gene expression analysis using FIBexDB that includes gene clustering and analysis of coexpression networks. Our findings shed new light on the role of lectins in plant defense and development in general, as well as on the potential functions of specific proteins with lectin domains.

## 2. Results

### 2.1. Expression of Flax Genes for Lectins during Fungal Infection in Different Cultivars

To evaluate the behavior of lectin genes during pathogen attack, we used RNA-SEQ data for flax root tips treated with *F. oxysporum* [10] downloaded in the FIBexDB (Figure 1a). We found that in root tips, 304 genes with lectin domains were expressed (threshold of TGR (total gene read) value ≥ 16 at least in one sample). Ninety-five percent of the revealed genes were expressed in the flax stem tissues [6]. According to the cluster analysis performed using FIBexDB, this set of lectin genes was subdivided into three clusters (Figure 1b): 63 genes accounted for the cluster where genes in general showed down-regulation of expression in the root samples of all flax genotypes treated with *F. oxysporum* compared to controls (cluster 1). The expression of 216 genes was upregulated in all flax genotypes treated with *F. oxysporum* as compared to controls: 146 genes were upregulated in all infected plants independently of the flax genotype (cluster 2). Cluster 3 included 70 genes whose expression in the susceptible cultivars was less pronounced compared to the resistant cultivars and hybrids. The pattern of gene expression of one of two hybrids ((1x4), Figure 1a), differed from other cultivars: the gene expression during infection was downregulated and upregulated as compared to the non-treated flax plants to a lesser extent than in other cultivars (Figure 1b).

### 2.2. Cluster 1: Lectin Genes Downregulated in Flax Root Tips in Response to Fusarium oxysporum

We analyzed the lectin families in each cluster to which the lectin genes belong. In cluster 1, comprising lectin genes downregulated during the infection, the proportion of genes encoding lectins belonging to the Galactose-binding lectin family (Gal_lectin), the Galectin-like family, and the Calreticulin family was more abundant compared to the other two clusters (Figure 1c, Appendix A). Among the genes of cluster 1, the expression of which was drastically changed, the genes encoding proteins with the lectin domain galactose-binding lectin (PF02140, Gal_lectin) which also have galactosidase-domain were presented: two paralog genes for beta-galactosidase *BGAL3* (*Lus10041798*, *Lus10028348*), two paralog genes for *BGAL8* (*Lus10000271*, *Lus10036108*), *BGAL10* (*Lus10003343*), *BGAL9* (*Lus10014278*) and *BGAL1* (*Lus10015625*) (Figure 2a, Appendix A). All the listed galactosidases belong to group “a”. Among the genes coexpressed with *LusBGAL3* (*Lus10041798*) were the genes encoding proteins involved in cellulose and pectin biosynthesis (LusCESA3 (*Lus10012198*), *LusCESA6* (*Lus10002939*), *LusGAUT14* (*Lus10013189*), *LusGAUT13* (*Lus10030703*)) (Figure 2b, Appendix A). The genes for putative S-adenosyl-L-methionine-dependent methyltransferases (SAM_Metases) had a pattern of expression similar to *BGAL3*: six genes for SAM_Metases had a correlation coefficient *r* > 0.8.

Downregulation of genes involved in the cell wall organization or biogenesis was detected in response to *F. oxysporum* in flax cultivars [10]. The authors claimed that in the susceptible cultivars, downregulation was more pronounced. This conclusion was based on the Gene Ontology (GO) analysis of the top 100 downregulated genes. According to the analysis by FIBexDB, the expression of genes encoding proteins involved in the primary cell wall biosynthesis was downregulated during the infection in all analyzed cultivars in a similar manner (Figure 3).

In cluster 1, several malectin genes, the expression of which was significantly suppressed in the course of the infection, were found. Two malectin genes (*Lus10035954* and *Lus10025708*) encode the Di-glucose binding proteins with kinesin motor domains. We analyzed the list of coexpressed genes for *Lus10035954* (Appendix A) and found that most gene products were involved in cytoskeleton functioning (based on Gene Ontology analysis of Arabidopsis ortholog genes). The coexpression of malectin genes with the genes encoding the cyclin-dependent protein kinase regulator (*Lus10042739*, *Lus10016015*, *Lus10029700*, and *Lus1001226*1), ATP binding microtubule motor family protein (*Lus10027488*, *Lus10025304*, *Lus10033133*, *Lus10039243*, and *Lus10024429*) was revealed (Figure 2c, Appendix A).

To evaluate the behavior of lectin genes from this cluster not only in roots but in stem tissues during normal plant development, the list of genes from cluster 1 was used for clustering using both root and stem samples (Figure 2d). When using stem samples for clustering, cluster 1 was split into several groups. The genes with a high level of expression in control root samples and suppressed during the infection (subcluster a, Figure 2d) were not highly expressed in stem tissues under normal conditions. Cluster 1 was enriched with the genes for calreticulin (PF00262) compared to other clusters (Figure 1c), and namely, the genes encoding calreticulins were among those with the highest level of expression in the control root, the expression of which was suppressed under *F. oxysporum* infection: *Lus10010923* and *Lus10010924* encoding calreticulins (CRT1a, both homologous to AT1G56340), *Lus10032521* and *Lus10043021* encoding calnexins (CNX1, both homologous to AT5G61790). These genes were highly expressed in the stem tissue, but under normal conditions their expression in the root tips was higher as compared to most stem tissues, and during infection, their expression in the root dropped to the same level as in other tissues (Figure 4a). The expression of genes for calnexins in the xylem (sXYLa) was comparable to the level of expression in the control root samples. In addition, the genes encoding CNX1 were upregulated in stem tissues (phloem fibers and xylem) in the course of graviresponse (Figure 4a, black arrows). The calreticulin gene *Lus10032521* was coexpressed with other calreticulin genes (*Lus10010923*, *Lus10043021*), genes for cell wall bisosynthesis, and the gene for calmodulin 5 (CAM5, *Lus10037423*) (Figure 4b). We suppose that the expression of these calreticulin genes is activated in the meristematically active zone of the root tips since in the slightly longer root segments (green arrows), their expression decreased (Figure 4a, green arrows).

The genes downregulated in roots during the infection but which initially had a lower level of expression in the control root samples were expressed in stem tissues under normal conditions. More than half of the lectin genes with relatively constant expression (34 out of 59, [6]) were present in cluster 1 (Appendix A). Fourteen out of seventeen lectin genes reported as the genes with a high level of expression in the flax stem tissue samples with the primary cell wall (Appendix A, [6]) were present in cluster 1, and 10 of them formed subcluster d (Figure 2d).

### 2.3. Cluster 2: Lectin Genes Upregulated in Flax Root Tips in Response to F. oxysporum

The cluster 2 included genes encoding lectins that perform the defense function inherent to lectins. Most of the flax genes with lectin domains (70% of the expressed genes) were upregulated in the root tips under *F. oxysporum* infection and fell into clusters 2 and 3. The genes in cluster 2 that had the highest level of expression in the root tips after inoculation with *F. oxysporum* belong to the Hevein, Nictaba, CRA, and Malectin-like families (Figure 1c, Appendix A).

The genes encoding proteins with Hevein domains (lectin domain coupled with glycoside hydrolase domain in some chitin-binding proteins) were presented exclusively in cluster 2, and some of the members were significantly upregulated in the flax root samples inoculated with *F. oxysporum*. *Lus10028377* and *Lus10041830* (both homologous to *PR-3*, *AT3G12500*) were significantly upregulated (a hundred times) in the root tips during infection by *F. oxysporum* (Figure 5a). Members of the PR-3, PR-4, PR-8, and PR-11 protein families are endochitinases that can hydrolyze chitin from the fungal cell walls [21,22,23]. Coexpression of *Lus10028377* with the genes encoding PAMP-INDUCED SECRETED PEPTIDE 2 (*Lus10010255* (*r* = 0.998), *Lus10019310* (*r* = 0.996)), cytochrome P450 enzymes (*Lus10022762* (*r* = 0.997), *Lus10012879* (0.9924), *Lus10034230* (*r* = 0.99)), *Lus10025901* (*r* = 0.99)), short-chain dehydrogenase/reductase (SDR) family proteins (*Lus10032192* (*r* = 0.99), *Lus10021313* (*r* = 0.99)) was revealed using the FIBexDB options (Figure 5b, Appendix A). Expression of gene for the protein related to the class V chitinases (*Lus10019060*) was also upregulated during the infection. This gene is homologous to *AT4G19810* of *A. thaliana*, encoding the CRA family protein which is mainly devoid of chitinase activity [24]. Of note, the genes mentioned above were similarly upregulated under the infection in the root samples of all analyzed flax genotypes and had a low level of expression in other flax tissues (stems, leaves, and hypocotyls).

The genes related to the Nictaba family were abundant in cluster 2 as compared to the other two clusters (Figure 1c). The *Lus10031473* gene encoding lectin of the Nictaba family was significantly upregulated in the root tips during the fungal infection (Figure 5a). The analysis of genes coexpressed with *Lus10031473* revealed that 5 members of the same family (*Lus10031471*, *Lus10031472*, *Lus10015208*, *Lus10015209*, and *Lus10031484*) were in the network (*r* > 0.86) (Figure 5c, Appendix A). Of note, 7 genes encoding different members of the transcription factor family WRKY (*WRKY40*, *41*, *33*, and *26*) were coexpressed (*r* > 0.84) with *Lus10031473* (Figure 5c, Appendix A). All the mentioned Nictaba genes are coexpressed with *Lus10039911* (homolog of *AT1G19180* encoding JAZ1, a nuclear-localized protein involved in jasmonate signaling).

It should be noted that lectin genes which were significantly activated in the flax root tips after inoculation with *F. oxysporum* were expressed in the root tips at a definite level under normal conditions (for instance, *Lus10028377* (Hevein family), *Lus10000579* (GNA family), *Lus10023391* (GNA family)). While a number of lectin genes (*Lus10031471* (Nictaba family), *Lus10032944* (GNA family), *Lus10018602* (Legume family)) and other genes for defense proteins coexpressed with chitinase *Lus10028377* were not expressed (or weakly expressed) in the control samples, the expression of these genes was specifically upregulated during the infection. The latter group included genes encoding PAMP-INDUCED SECRETED PEPTIDE 2 (*Lus10010255* and *Lus10019310*), SDR (*Lus10032192* and *Lus10021311*), cytochrome P450 CYP71B23 (*Lus10029035*), CYP715A1 (*Lus10027480* and Lus10027481), ethylene response factor (*Lus10025430*) and others (Figure 5b, Appendix A).

Clusterization of the upregulated lectin genes from cluster 2 using the flax stem samples and the root samples revealed that lectin genes highly upregulated in roots during the infection had a low level of expression in the stem tissues (subclusters b and d, Figure 5d, Appendix A). Subclusters b and d differed in the initial level of expression in the control samples without treatment with *F. oxysporum*: in subcluster b, the level of expression of genes in the control samples was low (sometimes 0), while in subcluster d, the genes had a relatively high initial level of expression. The average level of expression of lectin genes in subcluster b increased from 120 to 1500 TGR, while in subcluster d it increased from 900 to 1900 (upregulation by 13 and 2 times, respectively). The genes which were upregulated to a lesser extent during the infection were upregulated in the mature xylem part of the stem (subcluster a, Figure 5d) and phloem fibers with tertiary cell wall (subcluster c, Figure 5d). Fifteen of twenty-eight lectin genes that were reported as the genes upregulated in the flax stem tissue samples with the secondary cell wall [6] were upregulated during the infection (Appendix A). Among the genes in subcluster b were the genes for members of the Nictaba family (*Lus10015208*, *Lus10015209*, *Lus10031472*, and *Lus10031473*). In general, lectin genes upregulated in the root tips during infection were not expressed in the stem tissues bearing the primary cell walls (Figure 5d). Seventeen of fifty-five lectin genes with a constant level of expression in the flax stem tissues [6] were upregulated after the Fusarium treatment and fell into different subclusters (Figure 5d, Appendix A).

Clusters of genes for members of the GNA family were mainly activated during the infection (clusters 2 and 3, Figure 1c). Only *Lus10000579* (GNA family) from cluster 2 had a high level of expression in the control plants (more than 10,000 TGR) and the level of its expression during the infection was 2–3 times increased compared to the control plants (more than 23,000 TGR).

### 2.4. Cluster 3: Genes for Lectins Essentially Upregulated in Cultivars Resistant to Fusarium oxysporum

Cluster 3 included lectin genes that were upregulated in flax root tips during the infection, but in the susceptible flax genotypes this upregulation was less pronounced compared to the resistant cultivars and their hybrids (Figure 1b and Figure 6, Appendix A). As follows from the heatmap, in hybrid #3896 x AP5 (1x4), the expression pattern of lectin genes differed from the resistant cultivars and hybrid Dakota x AP5 (1x3) and was closer to the susceptible ones (Figure 1b and Figure 6d).

In cluster 3, specific abundant *Lus10005395* and *Lus10005397* genes (*LuALL3* and *LuALL2*, respectively, [25]) encoding proteins belonging to the Amaranthin family were revealed. In the coexpression network with the Amaranthin gene were present the genes involved in auxin signaling, defense response and root development (Figure 6b, Appendix A), the *DMR6* genes encoding a putative 2-oxoglutarate (*Lus10000711* and *Lus10024882*), genes for 1-DEOXY-D-XYLULOSE 5-PHOSPHATE SYNTHASE 2 (DXS) (*Lus10015519*, *Lus10019992*, *Lus10015518*), the auxin-responsive GH3 family protein (DFL1, *Lus10018510*), and lectin of the GNA family (*Lus10039731*).

There are several lectin genes the expression of which is more activated in the resistant cultivars: *Lus10042940*, *Lus10041680*, and *Lus10039732* encoding GNA lectins, *Lus10014651*, *Lus10035627* encoding Legume family lectins, *Lus10015417*, *Lus10013994*, *Lus10019237*, and *Lus10004275* for the Malectin family, *Lus10010443*, *Lus10010528* for the Malectin-like family, *Lus10020673* for the Nictaba family (Figure 6c, Appendix A).

We performed an additional clustering of lectin genes from cluster 3, extending the list of samples. It was shown that lectin genes which were essentially upregulated during the infection in the root tips were not expressed in the stem tissues (subcluster c, Figure 6d, Appendix A). In subcluster c, lectin genes were expressed in the control root samples and were upregulated during the infection, while in subcluster d, lectin genes had a low level of expression in the control root samples and their level of expression significantly increased during the infection (Figure 6d). There were lectin genes upregulated in fibers with tertiary cell walls (subcluster b, Figure 6d) and in xylem tissue (subcluster a, Figure 6d). While the expression of these genes was only slightly activated during the infection, these lectins probably accumulated at a definite level in some tissues, such as roots, but the encoded products of these genes fully performed their basic functions in the corresponding tissues.

## 3. Discussion

During the analysis of the lectin gene expression in different flax tissues and organs, the expression of 85% of lectin genes was detected in all analyzed tissues, but at different levels. We can assume that there is a fine-tuning of the lectin gene expression that is associated with the development processes and may differ even for one tissue (organ) at slightly different stages of development, while common pools of lectins in young tissues are present. About 75% of all predicted flax lectin genes were expressed in the root tips (5 mm segment with a relatively simple set of tissues) during normal conditions or fungal infection, and about half of them were upregulated in response to the *F. oxysporum* infection independently of flax genotype and resistance to Fusarium wilt.

In the current work, we did not localize lectin gene expression in flax root samples. There are several types of cells with different functions: columella, epidermis, cortex parenchyma, endodermis, pericycle, and protoxylem. Despite the fact that the cortex parenchyma is abundant compared to other tissues, the input into the transcriptome profile from the meristematic active zone and developing xylem tissue should be considered.

### 3.1. Genes Encoding Lectins Potentially Involved in the Biosynthesis of Cell Compounds

About 20% of the lectin genes expressed in the root tips were downregulated during the fungal infection (Figure 1(cluster 1) and Figure 2). Some of them had initially high levels of expression in the control samples (subcluster a, Figure 2d). Some of them with lower levels of expression were also downregulated during the infection (subclusters b, c, d, and e, Figure 2d). The degree of suppression for these groups was comparable: on average, the gene expression was 1.5–2.8 fold downregulated (calculated as the average ratio of CTR/FOX for different subclusters). Genes belonging to the Calreticulin and Gal_lectin families were enriched in the cluster of genes downregulated during the infection.

Among the genes initially with a high level of expression were those encoding calreticulins. Arabidopsis calreticulins (CRTs) and calnexins (CNXs) are known as the glucose-binding lectins located in the endoplasmic reticulum [8], involved in the folding and subunit assembly of the majority of the Asn-linked glycoproteins [26]. CRTs were referred to as the proteins involved in the folding of other proteins [27] with stable gene expression levels in all analyzed flax tissues [6]. Their downregulation in the course of the infection, when all general biosynthetic processes are suppressed, is reasonable. Moreover, it was shown that the loss of function of CRT1a (AT1G56340) reduces plant susceptibility to the fungal pathogen *Verticillium longisporum* in both *Arabidopsis thaliana* and *Brassica napus* [28] and results in activation of the ethylene signaling pathway, which may contribute to the reduced susceptibility.

Under osmotic or other abiotic stresses, CNX showed a highly reducible accumulation in the developing soybean roots [29], but a negative correlation between the calreticulin gene expression and the defense reaction development to the fungal infection was not reported. Of note, the expression of plant *CRTs* was increased in response to gravistimulation [30]. A similar dynamics of the *CRT* expression was observed in gravistimulated flax tissues (Figure 4a, black arrows). It was shown that the transcripts coding for calreticulin and calmodulin were recruited into polyribosomes predominantly in the lower half of gravistimulated maize pulvini [30]. Of note, the gene for calmodulin 5 was in the network of analyzed flax calreticulins as well (Figure 4b). It was suggested that an increased synthesis of calreticulin and calmodulin after gravistimulation would affect the cellular Ca^2+^ homeostasis and might play a role in the capacity and sensitivity to Ca^2+^ signals during the development of graviresponse [30]. Obviously, intensive protein biosynthesis and related processes of protein folding control with the involvement of calreticulins are inherent in root tips (close to the root apical meristem), shoot apical meristem, and gravistimulated stem tissues, and are suppressed in the course of stress reactions caused by fungi.

Previously, it was reported that genes involved in cell wall organization or biogenesis were significantly downregulated in the root tips in response to *F. oxysporum* [10,31]. Our data confirmed this statement, but by the example of genes for proteins involved in the primary cell wall biosynthesis (CESAs, GAUTs, CSLCs, XXT1), we did not find differences between the susceptible and resistant flax cultivars (Figure 3a). According to our results, the susceptible and resistant cultivars differ in the expression patterns of the cell-wall related genes, such as the genes encoding hydroxyproline O-galactosyltransferases, mannosidases, glucan endo-1,3-beta-glucosidases, flavonoid glucosyltransferases, UDP-glucosyltransferases, and others (data not shown).

Most of the lectin genes referred to as potentially associated with the primary cell wall development [6] were suppressed during the fungal infection. Among the genes encoding proteins with lectin domains drastically downregulated during the infection in all flax genotypes, the *BGAL* genes were widely presented. Based on the list of genes from the coexpression network with the *BGAL3* (*Lus10041798*) and their Gene Ontology annotation (Figure 2b), we can assume that products of the *BGAL* genes with lectin domain are closely associated with the primary cell wall formation. Such associations can be related to the modifications of pectin or other hemicelluloses that compose the primary cell wall. For example, xyloglucan was determined as a substrate for Arabidopsis *BGAL10*. This *BGAL* activity was required for the structural features of xyloglucan, which in turn provided a suitable environment for interaction with cellulose microfibrils during the primary cell wall extension in young tissues [32]. Though *LusBGAL3* was predicted as an extracellular protein, it was closely coexpressed with the genes encoding enzymes involved in pectin biosynthesis: galacturonosyltransferases (GAUTs) and (SAM)-dependent methyltransferases (SAM_Metases) (Figure 2b). SAM_Metases catalyze the transfer of a methyl group to an acceptor molecule. This family of plant methyltransferases contains enzymes that act on a variety of substrates, including salicylic acid (SA), jasmonic acid (JA), and 7-methylxanthine [33]. Involvement of a putative Arabidopsis S-adenosyl-L-methionine (SAM)-dependent methyltransferase, termed QUASIMODO 3 (QUA3, At4g00740), in methylesterification of the pectin homogalacturonan (HG) was supposed [34]. Flax ortholog of *AtQUA3* (*Lus10000808*) was presented in the coexpression network of *LusCNX1* (Figure 4b); *Lus10037101* (homolog of *AT1G13860*, encoding QUASIMODO2 LIKE 1) was found in the coexpression network for *LusBGAL3* (Appendix A). The degree of methylesterification of pectin affects the adhesive properties of pectin [35]. The association of *LusBGAL* gene expression and HG biosynthesis is probably not direct, but we can assume that *BGALs* with lectin domains are closely associated with the primary cell wall biosynthesis, which is suppressed during fungal infection.

During the infection, the expression of genes encoding malectins with an additional kinesin domain was downregulated (Figure 2a). According to the list of coexpressed genes, these malectins are involved in the microtubule cytoskeleton organization (Figure 2c). The orthologous gene in Arabidopsis *At2g22610* encodes kinesin MDKIN2, which, as was suggested, may be related to its cargo or protein binding ability and has diverged functions at different points in the cell cycle and in different subcellular locations. GUS staining for MDKIN2 revealed expression in diverse plant tissues but predominantly in the vasculature and dividing tissues such as growing leaves and root tips [36]. The carbohydrate specificity of the malectin domain in these proteins needs to be deciphered. It is suggested that if the MDKIN2 malectin domain can similarly bind to polysaccharides, its localization during cytokinesis may indicate a function in the organization of cell wall components [36]. This assumption is supported by the revealed coexpression with the flax malectin gene (*Lus10035954*) of the gene encoding CSLD5.1 (*Lus10010024*, ortholog of AT1G02730) (Appendix A). For *CSLD5*, the expression has been shown to be cell cycle dependent and appears to be involved in the phragmoplast formation in Arabidopsis [37].

### 3.2. The Avant-Garde of Defense Flax Lectins in Response to Fusarium oxysporum

Absolute champions among genes whose expression was significantly upregulated in the flax root tips during the fungal infection were those belonging to the Hevein and CRA families: the level of expression increased a hundred times compared to the control samples. These proteins possess a chitin-binding domain along with chitinase activity and cannot be considered as “true” lectins [38]. Nevertheless, these proteins with lectin domains play an important role in plant defense and are at the front of the battle with pathogens. For the Hevein-like antimicrobial peptides, binding to chitin is supposed to be vital for their antifungal activity, although the exact mechanism of their action remains unknown [39]. Activation of the oxidative and reductive processes during the infection is a common reaction in plants and there are plenty of known participants, including cytochromes p450 and short-chain dehydrogenase/reductases [40]. In plants, cytochromes P450 are closely related to the biosynthesis of phytoalexins [41]. In total, 31 genes for the cytochrome p450 superfamily belong to the following different families: CYP71, CYP76, CYP81, CYP82, CYP84 (all of them are in clan CYP71 [41]) coexpressed with *Lus10028377* (*r* > 0.95) (Appendix A). Fifteen *SDR* genes (short-chain dehydrogenase/reductase (SDR) family protein, SDR2 and SDR5) were found to be coexpressed with *Lus10028377* (coefficient of coexpression *r* > 0.94). We can assume that different members of the cytochrome p450 superfamily and SDR act on different steps of diterpene phytoalexin metabolism, as was shown for rice [42], but whether chitinases are directly involved in this process remains unknown. It was shown that chitin oligosaccharides from the fungal cell walls induce the momilactone cluster genes in rice, one of two gene clusters involved in the biosynthesis of rice diterpene phytoalexins [42].

Among the lectins whose gene expression was upregulated under the Fusarium infection, proteins with the Nictaba domain were present (Figure 5a). The first carbohydrate-binding protein with the Nictaba domain was observed in the leaves of *Nicotiana tabacum* plants after treatment with jasmonates or insect herbivory [43]. Later, the interaction of the Nictaba lectin with different core histones was shown within the nucleus of the plant cell through their O-GlcNAc modification [44]. Interestingly, all of the Nictaba genes were found to be coexpressed with the *Lus10039911* gene, which encodes JASMONATE-ZIM-DOMAIN PROTEIN 1 (JAZ1) and a number of genes for WRKY transcription factors (WRKY40, 41, 33, and 26, Appendix A). It was shown that some members of WRKY (WRKY40, 57, and 75) are able to activate expression of the JAZ genes, which are known as JA-induced repressors of the JA signaling pathway [45,46,47]. The WRKY transcription factors are widely known for their vital role in the plant’s response to both external and internal stimuli, including the response to the fungal pathogen *F. oxysporum* [48]. It was shown that WRKY33 may function as a downstream component of the MPK4-mediated signaling pathway and contribute to the repression of SA-dependent disease resistance [49]. Specific upregulation of the Nictaba genes in the root tips and mature xylem under normal conditions (Figure 5c) gives us reason to suggest that flax Nictaba genes for proteins with F-box domains (*Lus10015208*, *Lus10015209*, *Lus10031473*, *Lus10031471*, all homologous to At1G56250), are specifically expressed in the vascular system. For Arabidopsis plants, it was shown that the *At1G56250* gene product is a VBF (VIP1-binding F-box) protein. VBF expression is known to be induced by *Agrobacterium* and facilitate tumor formation, suggesting that the host factors that the pathogen uses for infection also include those that the plant initially produces to defend itself from the very same infection [50].

Xylem tissue is at the “forefront” of fungal attack: after penetration of the root epidermis, hyphae progress intercellularly via the root cortical cells until they enter the xylem tissue, where wilt processes develop [51]. In this case, it could be reasonable to suggest that developing xylem tissue in the root tips accumulates quite a number of defense proteins, including lectins and other substances, in order to be ready in advance to face the possibility of a pathogen attack. We can assume that the xylem-specific lectins are more likely to be involved in the defense processes and modification of various substrates than in biosynthetic processes. Although in the root tips, the elements of xylem only begin their development, the first thickened protoxylem cells already appear (Figure 1a).

Some genes encoding defense proteins, including lectins, have a definite level of expression in the tissues under normal conditions. A pathogen attack increases the transcript abundance, while the expression of a number of genes is strongly induced during the infection. The lectin genes activated from a low level (TGR < 16) were mostly from the GNA and Legume families. *Lus10000579*, encoding a GNA family member, had a high level of expression in the root tips and increased it during the infection; it is a homolog to *AT1g78850* (*AtGAL1*), which encodes a curculin-like (mannose-binding) lectin family protein. It was shown that the Arabidopsis cells in a suspension culture accumulate this lectin as well as endochitinase AT3g12500 (homolog for *Lus1002837*7, which was described above) in response to a chitosan elicitor in the cell wall fraction [52]. Under normal conditions, *Lus10000579* had the highest level of expression in hypocotyls and root tips; this gene probably plays an important role in the early stages of development, being susceptible to fungal infection.

### 3.3. Lectins That Might Determine the Resistance of Flax to F. oxysporum

In the course of clusterization of the flax lectin genes in different stem tissues and root samples, a group of lectin genes whose expression differed in the susceptible and resistant flax cultivars was revealed (Figure 1b and Figure 6). The genes encoding amaranthins (*Lus10005395* (*LuALL3*), and *Lus10005397* (*LuALL2*)) had the highest level of expression in all flax genotypes, even in the control samples, and were upregulated under the fungal infection, this upregulation being more pronounced for the resistant cultivars and their hybrids. Transcripts of both these genes significantly increased in abundance in the etiolated flax seedlings following the methyl jasmonate (MeJA) treatment, but not by SA [25]. In the root tips and seedlings, the *LuALL2* and *LuALL3* transcripts were generally abundant in untreated tissues as well [25]. It is well known that MeJA is involved in the activation of a disease-related defense system. It was shown that the signaling pathway of the initially anti-stress plant hormone MeJA plays both a positive and a negative role in the resistance of *A. thaliana* to *F. oxysporum*. This is due to the fact that pathogens reprogram the physiological functions of the host to increase susceptibility to infection [53]. Indeed, *Lus10004990* (homolog of *AT4G36920*) encoding an ethylene responsive APETALA2 (AP2)-domain transcription factor being also a jasmonate-responsive gene [54] was upregulated in the infected root samples and activation in the resistant cultivars was more pronounced (data not shown). Interestingly, flax has one of the largest families of amaranthin-like lectins among other plant species [25]. Many taxonomically distant plant species, including Arabidopsis, do not have amaranthin-like lectins in their genomes [6,55].

We found that amaranthin genes were in coexpression with 1-DEOXY-D-XYLULOSE 5-PHOSPHATE SYNTHASE 1 (*DXS1*) (Figure 6b). The DXS1 is associated with the production of terpenoids (MEP (non-mevalonate) pathway) [56,57]. It has been shown that in flax, the non-mevalonate pathway is strongly activated at the beginning of the *F. oxysporum* infection and a redirection of metabolites towards abscisic acid (ABA) synthesis takes place [14]. Thus, lectins from both these groups may be important for the investigation due to their involvement in the resistance of flax cultivars. The most promising of them (based on the dynamics of expression) are listed above. The hypothetical involvement of proteins with lectin domains in plant defense is summarized in Figure 7.

## 4. Materials and Methods

For the analysis of lectin gene expression, RNA-SEQ data obtained by Dmitriev et al. [10] and uploaded to FIBexDB [11] was used: the transcriptome data of two resistant (Dakota and #3896) and two susceptible (AP5 and TOST) flax cultivars with respect to Fusarium wilt, as well as their two cross-breeds (Dakota/AP5, and #3896/AP5) (Figure 1a). There were a total of six pairs for comparison (control and treated with *F. oxysporum* plants). Figure 1a shows simplified sample names designating different flax cultivars in the experiment with *F. oxysporum* [10]. Seven-day-old flax plants were inoculated with *Fusarium oxysporum* f. sp. *lini* (pathogenic isolate #39) spores or with sterile water (control) [10]. For subsequent RNA-SEQ analysis, only the root tips of seedlings (approx. 5 mm in length) after 48 h of fungal treatment were used [10]; such time points afforded consideration of the early local responses. We also observed the cross-section of flax root tips (7 day-old seedlings, Mogilevsky cultivar, root segments approx. 5–8 mm in length) to establish what type of cells comprised the root samples at the analyzed stage (Figure 1a). Root segments were put in 3% (*w*/*w*) low-melting point agarose blocks and cross-sections (70 µm thick) were prepared using a Leica VT 1000S (Leica Biosystems) vibratome. Sections were incubated in Calcofluor White 1:100 solution in PBS and observed using an Axio Scope A1 microscope (Carl Zeiss MicroImaging GmbH; Jena, Germany).

The gene expression analysis was performed in silico using tool options within FIBexDB ([11], Figure 8). All data uploaded to FIBexDB was normalized. The number of reads for each gene was calculated as total gene read (TGR) counts. DESeq’s estimate Size Factors and estimate Dispersions functions (with default options) were used to obtain normalization factors for each sample and to normalize TGR counts [58]. The normalization was applied to all samples simultaneously to ensure that the expression values were comparable across samples. Genes with averaged normalized TGR ≥ 16 in at least one sample were considered as expressed according to the recommendations of the sequencing quality control project [59]. The list of flax lectins and their annotations was taken from [6] (Appendix A, and gene lists from Tables 2–5 of the main text in [6]). The list of analyzed genes was put into the “search by multiple queries” and then we chose samples for analysis. A clustering analysis was performed after choosing genes and samples. The k-means (Cluster3) method and the Pearson correlation were used as parameters for clustering. K-means is a distance-based algorithm, where the distances are calculated to assign a point to a cluster. In k-means, each cluster is associated with a centroid. Then, the results were extracted as tab-delimited text for further processing in Excel^®^. To analyze coexpressed genes, a gene page for each individual flax gene was used, and the data was downloaded as tab-delimited text. FIBexDB and the Appendix A of the related paper [12] contain a detailed sample description, a tutorial file, the NCBI Acc No. of the used RNA-SEQ projects, and references. Gene ontology (GO) enrichment analysis for orthologous genes of Arabidopsis was performed using http://geneontology.org/ (accessed on 8 December 2021). For Arabidopsis gene annotation, TAIR [60] and ProtAnnDB (http://www.polebio.lrsv.ups-tlse.fr/ProtAnnDB/index.php, accessed on 8 December 2021 [61]) were used.

## 5. Conclusions

FIBexDB is a useful and powerful tool for analyzing not only the participants involved in fiber biogenesis (as intended), but also the genes whose products, such as lectins, may be involved in normal tissue development and stress response to the *F. oxysporum* infection. Using FIBexDB tools, we could not only systemize lectin groups and suggest their functions, but reveal the changes in the lectin gene patterns inherent in different flax genotypes which varied in their resistance to the Fusarium wilt and identify lectins that might determine flax resistance. A wide panel of transcriptome samples allowed us to analyze the expression of genes encoding proteins with lectin domains in stress conditions (under *F. oxysporum* infection) and during normal plant development in different flax stem tissues that differed in their forming of cell wall types. These panel and FIBexDB options proved to be very convenient and efficient for the cluster analysis and the coexpression network analysis.

We found that about 75% of all predicted lectin genes are expressed in the root tips under normal conditions or under the *F. oxysporum* treatment. Most of the lectin genes were upregulated during the infection, while the expression of 20% of genes was suppressed. General biosynthetic processes, including biosynthesis of proteins and components of the primary cell wall, were downregulated under *F. oxysporum*, as well as gene expression of lectins that are potentially associated with the primary cell wall functioning or involved in protein folding (such as the Gal_lectin family, Calreticulin family, and Malectin family).

The genes encoding lectins of the Hevein, CRA, Nictaba, and GNA families were activated during infection in the flax root tips. Activation of expression of the Nictaba genes could be associated with cross-talk of the SA and JA signaling pathways. Among genes encoding lectins that could determine resistance to Fusarium wilt, the genes for amaranthins were revealed, the expression of which was MeJA-inducible in flax seedlings [25].

We believe that the results and algorithms presented in the current work will be useful for researchers engaged in plant lectin investigation and will allow them to establish the function of lectins and their relationship with all vital processes such as defense, protein biosynthesis, and cell wall functioning.

## Figures and Tables

**Figure 1 plants-11-00163-f001:**
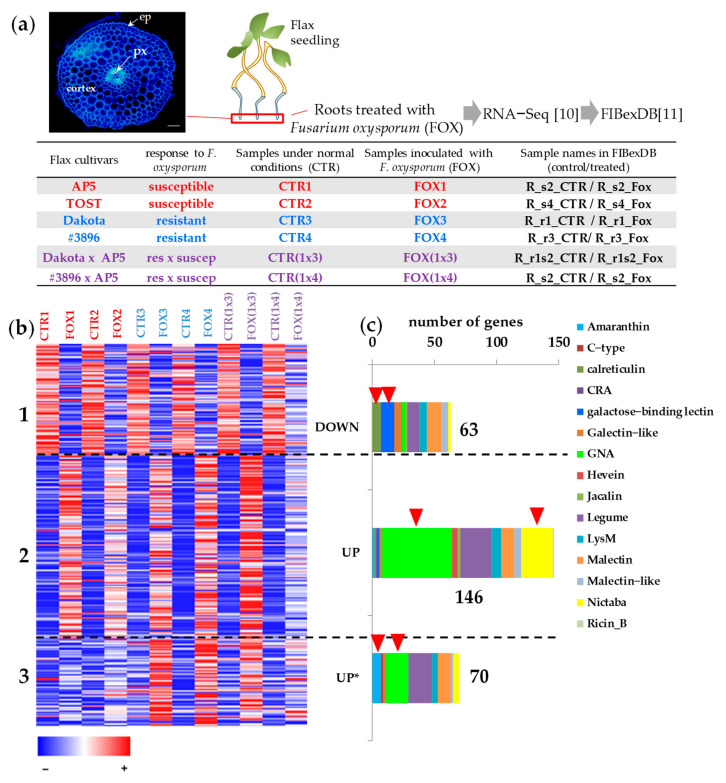
(**a**) Sample description (based on [10,11]). Cross-section of flax root tips (stained with calcofluor), ep—epidermis, px—protoxylem, bar 100 µm. (**b**) Cluster analysis of 304 genes expressed in flax root tips of different genotypes treated with *Fusarium oxysporum*. A heatmap was built in FiBexDB. Options for clustering: the k-means (3 clusters) method, a distance measured by the Pearson correlation, local (auto) scale. The color gradient of the red-blue heatmap reflects TGR values from minimum (blue) to maximum (red) in the row. CTR1,2,3,4,(1x3),(1x4)—untreated root tips of different flax genotypes, FOX1,2,3,4,(1x3),(1x4)—root tips of different flax genotypes treated with *F. oxysporum* [10]. *—genes for lectins upregulated in cultivars resistant to *F. oxysporum.* (**c**) Members of different lectin gene families in the analyzed clusters. Red arrows mark family members which were abundant in the analyzed clusters.

**Figure 2 plants-11-00163-f002:**
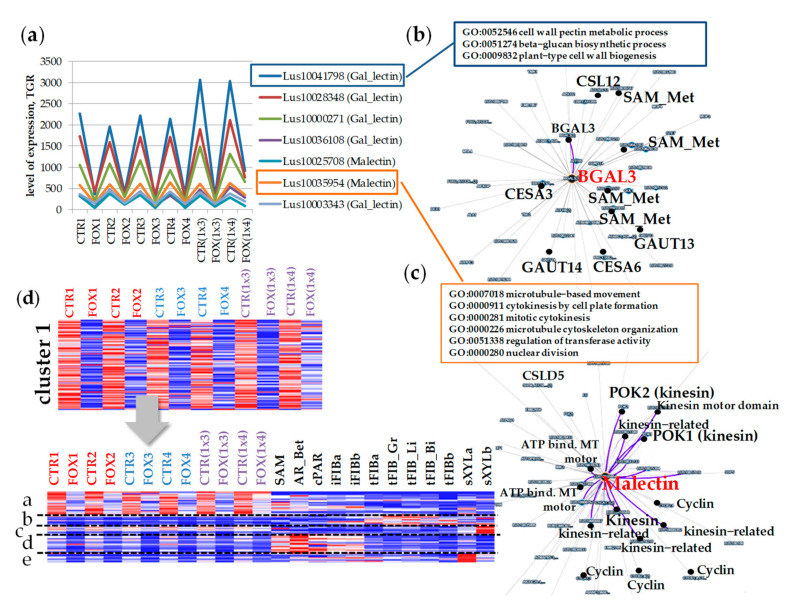
(**a**) The expression of several lectin genes mostly suppressed in the root tips during *Fusarium oxysporum* infection in all analyzed cultivars. (**b**) The coexpression network for *LusBGAL3*, its coexpressed genes, and their gene ontology (GO, for orthologous genes in Arabidopsis). (**c**) The coexpression network for *Lus10035954* (Malectin family), its coexpressed genes, and their gene ontology (GO, for orthologous genes in Arabidopsis). (**d**) The cluster analysis of 63 genes downregulated after inoculation with *F. oxysporum* for the root and stem samples. Options for clustering: the k-means (5 clusters) method, a distance measured by the Pearson correlation, local (auto) scale. The color gradient of the red-blue heatmap reflects TGR values from minimum (blue) to maximum (red) in the row. CTR1,2,3,4,(1x3),(1x4)—untreated root tips of different flax genotypes, FOX1,2,3,4,(1x3),(1x4)—the root tips of different flax genotypes treated with *F. oxysporum* [10]. SAM—shoot apical meristem [15]; AR_Bet—apical region (unpublished), cPAR—parenchyma cells, iFIB—the intrusively growing phloem fibers [16,17]; tFIBa, tFIB_Gr, tFIB_Li, tFIB_Bi, tFIBb—the isolated phloem fibers with tertiary cell wall (different cultivars, [17,18,19]; sXYLa, sXYLb—xylem part of the stem enriched with secondary cell wall [17,18]. The detailed sample descriptions and references are presented in FIBexDB and the Appendix A of the related paper [12].

**Figure 3 plants-11-00163-f003:**
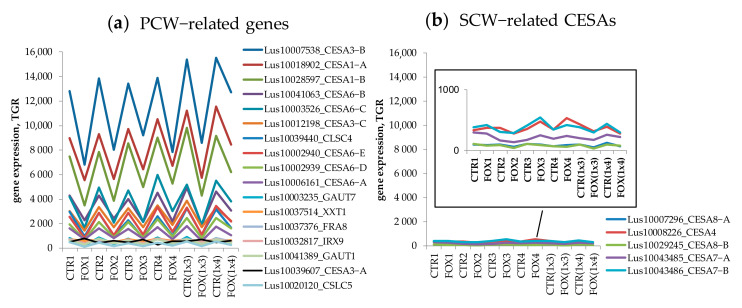
The expression of genes encoding cell-wall related biosynthetic enzymes during the infection by *Fusarium oxysporum*. (**a**) The expression of genes encoding products involved in primary cell wall (PCW) biosynthesis. (**b**) The expression of genes encoding cellulose synthases (CESAs) involved in the secondary cell wall (SCW) formation.

**Figure 4 plants-11-00163-f004:**
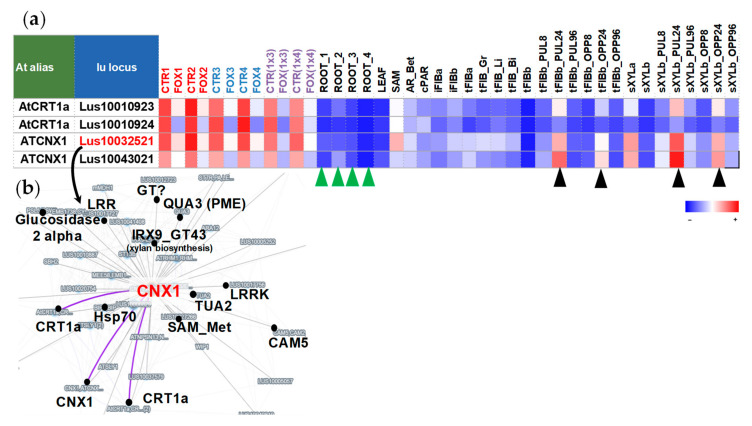
(**a**) The expression of genes for the Calreticulin (PF00262) family (calreticulins and calnexins) in different flax tissues. Black arrows mark stem samples (phloem fibers and xylem part) after gravistimulation [6]. Green arrows mark control root samples from an experiment with aluminum treatment [20]. The detailed sample descriptions and references are presented in FIBexDB and the Appendix A of the related paper [12]. The color gradient of the red-blue heatmap reflects TGR values from minimum (blue) to maximum (red) in the row. (**b**) The coexpression network of calreticulin *LusCNX1* (*Lus10032521*).

**Figure 5 plants-11-00163-f005:**
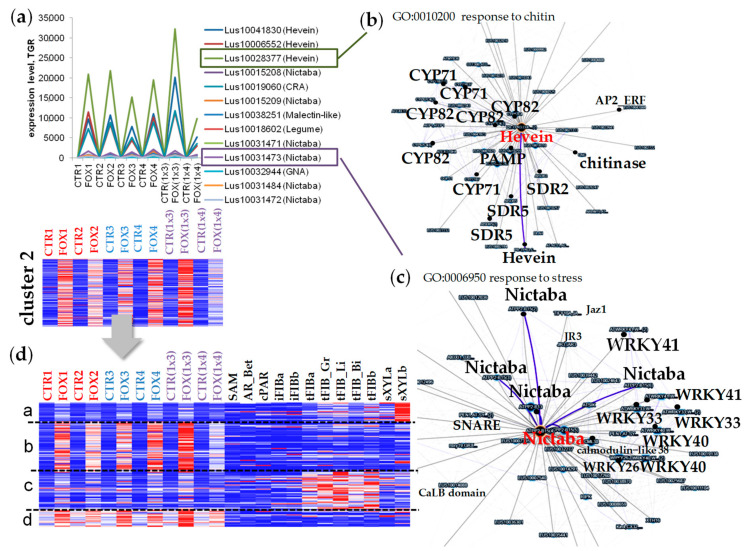
(**a**) The expression of several lectin genes mostly activated in the root tips during *Fusarium oxysporum* infection in all analyzed cultivars. (**b**) The coexpression network for *Lus10028377* (Hevein family), its coexpressed genes, and their gene ontology (GO, for orthologous genes in Arabidopsis). (**c**) The coexpression network for *Lus10031473* (Nictaba family), its coexpressed genes, and their gene ontology (GO, for orthologous genes in Arabidopsis). (**d**) The cluster analysis of 146 genes upregulated in the root tips after inoculation with *F. oxysporum* for root and stem samples. Options for clustering: the k-means (4 clusters) method, a distance measured by the Pearson correlation, local (auto) scale. The color gradient of the red-blue heatmap reflects TGR values from minimum (blue) to maximum (red) in the row. CTR1,2,3,4,(1x3),(1x4)—untreated root tips of various flax genotypes; FOX1,2,3,4,(1x3),(1x4)—the treated root tips of various flax genotypes [10]. SAM—shoot apical meristem [15]; AR_Bet—apical region (unpublished), cPAR—parenchyma cells, iFIB—the intrusively growing phloem fibers [16,17]; tFIBa, tFIB_Gr, tFIB_Li, tFIB_Bi, tFIBb—the isolated phloem fibers with tertiary cell wall (different cultivars, [17,18,19]; sXYLa, sXYLb—xylem part of the stem enriched with secondary cell wall [17,18]. The detailed sample descriptions and references are presented in FIBexDB and in the Appendix A of the related paper [12].

**Figure 6 plants-11-00163-f006:**
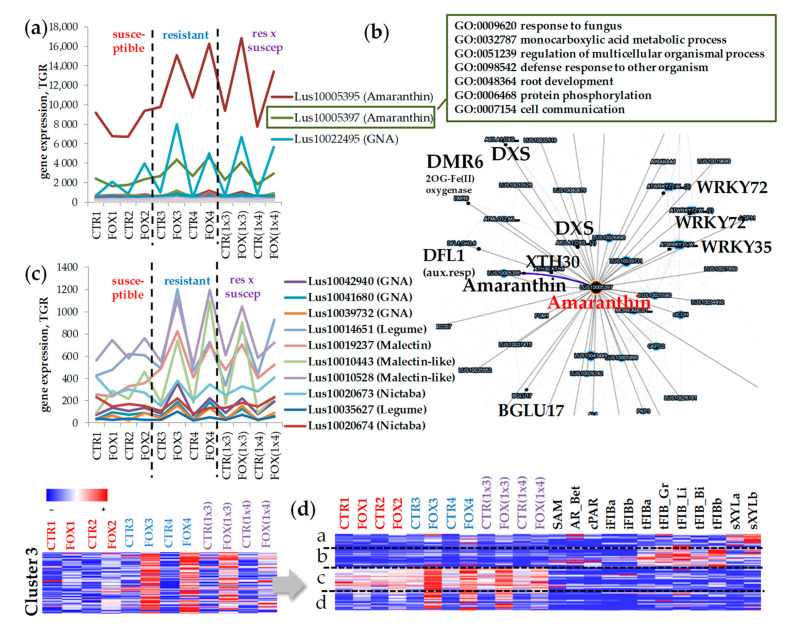
(**a**) The expression of lectin genes upregulated in the root tips during *Fusarium oxysporum* infection to varying degrees in different flax genotypes. (**b**) The coexpression network for *Lus10005397* (Amaranthin family), its co-expressed genes, and their gene ontology (GO, for orthologous genes in Arabidopsis). (**c**) The expression profile of lectin genes expression of which is more activated in the resistant cultivars. (**d**) Cluster analysis of 70 genes upregulated in root tips of susceptible cultivars to a lesser extent as compared to resistant cultivars after inoculation with *F. oxysporum* for root and stem samples. Options for clustering: k-means (4 clusters) method, a distance measured by Pearson correlation, local (auto) scale. The color gradient of the red-blue heatmap reflects TGR values from minimum (blue) to maximum (red) in the row. CTR1,2,3,4,(1x3),(1x4)—untreated root tips of various flax genotypes; FOX1,2,3,4,(1x3),(1x4)—the treated root tips of various flax genotypes [10]. SAM—shoot apical meristem [15]; AR_Bet—apical region (unpublished), cPAR—parenchyma cells, iFIB—the intrusively growing phloem fibers [16,17]; tFIBa, tFIB_Gr, tFIB_Li, tFIB_Bi, tFIBb—the isolated phloem fibers with tertiary cell wall (different cultivars, [17,18,19]; sXYLa, sXYLb—xylem part of the stem enriched with secondary cell wall [17,18]. The detailed sample descriptions and references are presented in FIBexDB and in the Appendix A of the related paper [12].

**Figure 7 plants-11-00163-f007:**
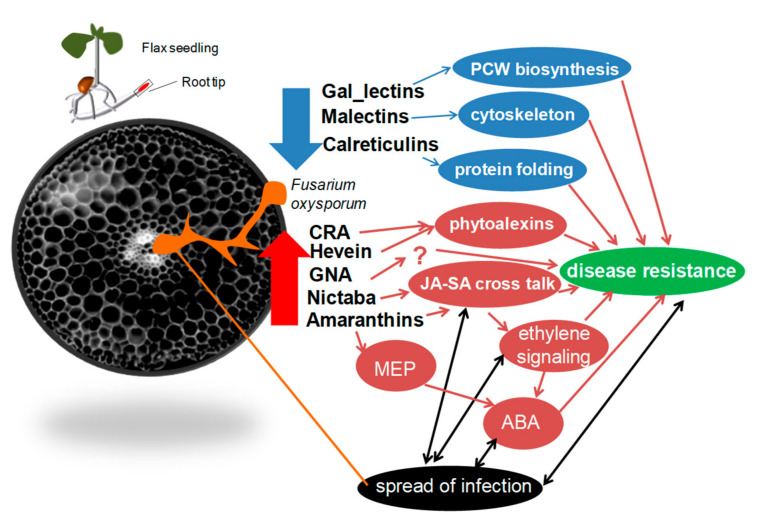
The hypothetical involvement of proteins with lectin domains in plant defense. The lectin gene whose expression changed drastically during the *Fusarium oxysporum* infection and their potential association with the general processes are presented. The downregulated and upregulated pathways are given in blue and red backgrounds, respectively. Lectins are part of a comprehensive network of plant-pathogen interactions that includes cross-talk between plant resistance and pathogen spread. SA—salicylic acid, JA—jasmonic acid, ABA—abscisic acid, MEP—non-mevalonate pathway.

**Figure 8 plants-11-00163-f008:**
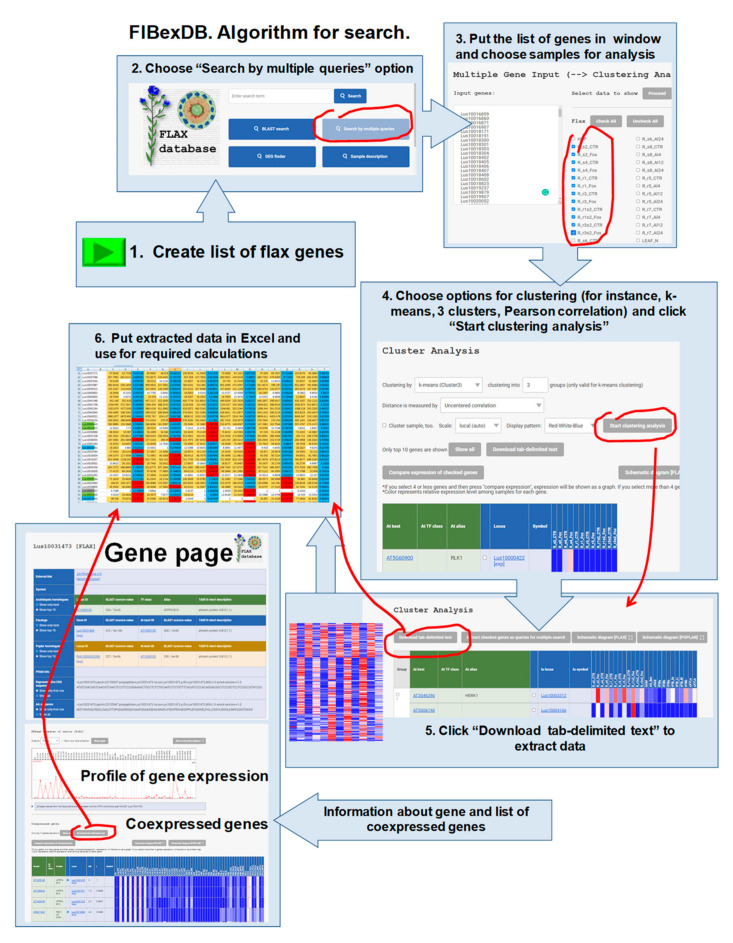
How to use FIBexDB for the analysis of expression data.

## Data Availability

https://ssl.cres-t.org/fibex/ (accessed on 8 December 2021).

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
