# Peer review of "Using FIBexDB for In-Depth Analysis of Flax Lectin Gene Expression in Response to *Fusarium oxysporum* Infection"

_plants, 2022, doi:10.3390/plants11020163_

Round 1
Reviewer 1 Report
Basically, I think it is quite a remarkable piece of work that you did and I suggest this manuscript could get published. Flax is an important and widespread in Europe as a commercial crop. Fusarium wilt has been considered as a major limiting factor in flax production and the use of available resistant varieties is the main strategy aimed to manage the problem. In the current manuscript authors investigated the role of lectin-domain containing proteins in flax response to Fusarium oxysporum by using specialized database named FIBexDB. This approach allowed them to reveal main lectin families participating in plant defense and to define genes specific to resistant cultivars that might determine their unsusceptible to fungal invasion. Overall, the MS is well written, the experiments were well performed, the data is of high quality and well presented. I would recommend it for publication after the authors address the following issues.
-The title of this manuscript is not specific and does not properly reflect the content. Authors studied the lectin gene expression under Fusarium oxysporum treatment and it must be defined clearly in title.
-In Introduction, please add some specific information concerning defense function of lectins, including their classification and mechanism of action. It is also not obvious why it is important to investigate these proteins in flax, please, point it out in this section.
-Fig.1. Closing bracket is missing (line 77).
-Fig.2, 5, and 6. Please, add the word “cluster” before number in (d).
-I think that validation of gene expression through quantitative PCR of the genes belonging to the Hevein and CRA families as well as those that might determine the resistance of flax to Fusarium could potentially enhance the reliability of the obtained results.
-The Conclusion section can be considerably reduced. The second and the third paragraph are more suitable for Discussion.
Reviewer 2 Report
The authors have focused on in-depth analysis of lectin genes in flax root tips of susceptible and resistant cultivars, exposed to Fusarium oxysporum for 48 hours. They used the expression data from their previous experiments (6), which also formed the basis of the database FIBexDB; with the aim to point out the possibilities of this database. Maybe that's why the Introduction is poor, and the Discussion is wide and sometimes overlaps with the Results.
Figure 1a - the same Figure is also on the FIBexDB database web page, and it would be better to put it right in the legend, even if it is the work of the authors.
Figure 1a (calcofluor staining), it is not clear for me, why this type of the experiment was performed; what this Figure should show, when the authors analysed resistant and sensitive flax cultivars.
The section Materials and Methods should be reorganized, in this style it is a bit confusing. Although the authors used data from previous experiments, it is necessary to clearly state the conditions under the stress experiments were performed (for the future correct use of presented conclusions).
Reviewer 3 Report
This manuscript demonstrates the utility of FIBexDB using the flax lectin gene expression as an example. I have the following questions and suggestions:
- More information related to the FIBexDB should be introduced early on. Is FIBexDB constructed by the same research group? This is missing in the abstract. Currently in the abstract it seems that a publicly available tool/database was used. It can be mentioned that previously they have built a database and in the current manuscript they are demonstrating the utility of this database - if this is the main goal it should be clearly stated.
- In defining the clusters, particularly the genes in Figure 1, have they used differentially expressed gene analyses? How were the up/down regulation defined? Were these statistically significant?
- In the methods, it is mentioned that "The list of analyzed genes was put into the "search by multiple queries" and then we chose samples for analysis. A clustering analysis was performed after choosing genes and samples." The selection of the genes is later discussed in the last paragraph. Reorganization of this section would be helpful for better understanding.
- In the methods, the choice of the clustering parameters should be discussed further. For example what does k-means (Cluster3) method mean? Have the authors tried different variables in the k-means clustering such as number of clusters, distance measure, multiple random centroids as nstart? If not, what would be the impact of changing these variables on the results, conclusion? Have they considered other clustering methods, it seems hierarchical clustering is also available in FIBexDB?
- The authors mention that "Gene ontology (GO) Enrichment Analysis was performed using http://geneontology.org/." Which organism was selected and what cutoffs were used for significant enrichment?
- In Figure 1b and other heatmaps, the legend is missing. What does the color depict, is that normalized counts? Which normalization method was used for the RNA-sequencing data?
- What was the interpretation from Figure 3b? The y-axis seems to be the same as Figure 3a, however cellulose synthase genes involved in secondary cell wall were expressed lower than the primary cell wall biosynthesis genes. The cellulose synthase genes pattern could be shown by adjusting the y-axis on the Figure 3b.
- How were the thresholds on TGR determined? For example: threshold of TGR value ≥ 16 at least in one sample. Also, TGR unit needs to be spelled out in the first usage.
- How was gene co-expression defined throughout the manuscript?
Round 2
Reviewer 2 Report
The authors responded sufficiently to my comments. The improved MS is suitable for publishing.
Reviewer 3 Report
I would like to thank the authors for addressing all my comments. Also, the manuscript has been improved.